# Ginsenosides Restore Lipid and Redox Homeostasis in Mice with Intrahepatic Cholestasis through SIRT1/AMPK Pathways

**DOI:** 10.3390/nu14193938

**Published:** 2022-09-22

**Authors:** Guodong Li, Yanjiao Xu, Qianyan Gao, Sheng Guo, Yue Zu, Ximin Wang, Congyi Wang, Chengliang Zhang, Dong Liu

**Affiliations:** 1Department of Pharmacy, Tongji Hospital Affiliated Tongji Medical College, Huazhong University of Science and Technology, Wuhan 430030, China; 2Department of Neurology, The First Affiliated Hospital of Xinxiang Medical University, Weihui 453199, China; 3The Center for Biomedical Research, Tongji Hospital, Tongji Medical College, Huazhong University of Science and Technology, Wuhan 430030, China

**Keywords:** intrahepatic cholestasis (IC), ginsenosides (GS), lipid metabolism, antioxidant

## Abstract

Intrahepatic cholestasis (IC) occurs when the liver and systemic circulation accumulate bile components, which can then lead to lipid metabolism disorders and oxidative damage. Ginsenosides (GS) are pharmacologically active plant products derived from ginseng that possesses lipid-regulation and antioxidation activities. The purpose of this study was to evaluate the possible protective effects of ginsenosides (GS) on lipid homeostasis disorder and oxidative stress in mice with alpha-naphthylisothiocyanate (ANIT)-induced IC and to investigate the underlying mechanisms. A comprehensive strategy via incorporating pharmacodynamics and molecular biology technology was adopted to investigate the therapeutic mechanisms of GS in ANIT-induced mice liver injury. The effects of GS on cholestasis were studied in mice that had been exposed to ANIT-induced cholestasis. The human HepG2 cell line was then used in vitro to investigate the molecular mechanisms by which GS might improve IC. The gene silencing experiment and liver-specific sirtuin-1 (SIRT1) knockout (SIRT1^LKO^) mice were used to further elucidate the mechanisms. The general physical indicators were assessed, and biological samples were collected for serum biochemical indexes, lipid metabolism, and oxidative stress-related indicators. Quantitative PCR and H&E staining were used for molecular and pathological analysis. The altered expression levels of key pathway proteins (Sirt1, *p*-AMPK, Nrf2) were validated by Western blotting. By modulating the AMPK protein expression, GS decreased hepatic lipogenesis, and increased fatty acid β-oxidation and lipoprotein lipolysis, thereby improving lipid homeostasis in IC mice. Furthermore, GS reduced ANIT-triggered oxidative damage by enhancing Nrf2 and its downstream target levels. Notably, the protective results of GS were eliminated by SIRT1 shRNA in vitro and SIRT1^LKO^ mice in vivo. GS can restore the balance of the lipid metabolism and redox in the livers of ANIT-induced IC models via the SIRT1/AMPK signaling pathway, thus exerting a protective effect against ANIT-induced cholestatic liver injury.

## 1. Introduction

Intrahepatic cholestasis (IC) is a hepatocellular disease in which abnormal bile formation, secretion, and/or excretion occur in the liver [1]. IC manifests internally as liver lesions caused by the excessive buildup of bile acids, cholesterol, bilirubin, and other bile components in the liver and systemic circulation. Patients with IC exhibit jaundice, weakness, dark colored urine, and pruritus [2]. Untreated, it can lead to cirrhosis, portal hypertension, and liver failure [1].

Generally, IC is considered a pathological result of the dysfunction of the excretion and secretion of bile; however, more recent studies have found that IC leads to disordered lipid homeostasis and oxidative stress in the liver [3]. Clinically, intrahepatic cholestasis of pregnancy (ICP) is known to be connected with an abnormal metabolic profile, including dyslipidemia and glucose intolerance [4]. Recent studies have demonstrated that total cholesterol (TC) and low-density lipoprotein (LDL) concentrations were significantly raised, but high-density lipoprotein (HDL) is drastically reduced in ICP patients and in rat models of bile duct ligation (BDL) [5,6,7]. PBC patients were also found to have high cholesterol levels in the blood [8]. However, the mechanism of lipid metabolism disorders in IC remains unclear. In addition to these observed effects on the lipid metabolism, oxidative stress also extensively exists in IC and is closely connected to the dysfunction in the lipid metabolism. Dysfunctions in the lipid metabolism result in hepatic lipid aggregation, which promotes fatty acid β-oxidation and leads to the manufacturing of more reactive oxygen species (ROS) in metabolic disease of the liver [9]. Excessive ROS-induced oxidative damage aggravates lipid accumulation by interrupting mitochondrial roles and cutting down fatty acid oxidization [10,11]. In summary, lipid metabolism disorders and oxidative stress are of great significance for the occurrence and development of IC, but the underlying mechanism has not been clearly elucidated.

It is extremely difficult to develop new drugs that target cholestasis due to the complex etiology and mechanism of injury. Ursodeoxycholic acid (UDCA) is effective for treating primary biliary cholangitis (PBC), but more than 40 percent of patients do not respond well to it [12]. Obecholic acid (OCA) can considerably increase the survival rate of PBC patients who do not respond to or tolerate UDCA therapy, but it is associated with significant side effects such as pruritus [13]. Thus, the discovery of new targets and the development of new therapeutic approaches are necessary.

*Ginseng Radix et Rhizoma* is one of the most widely used herbs in East Asian countries. Ginsenosides (GS) are the most abundant active component of *Panax ginseng* and have shown excellent pharmacological effects in the treatment of many diseases, including obesity, diabetes, and atherosclerosis [14]. GS has been reported to ameliorate glycolipid metabolism and reduce triglyceride accumulation in obese mice [15,16]. Other studies have found that GS can alleviate oxidative stress levels in various liver diseases [17,18].

In previous studies, we observed a liver-protective effect of a GS treatment in IC rats [19]. However, the mechanism of the GS treatment was not elucidated, and at present, we hypothesize that GS may improve IC by regulating lipid metabolism and oxidative stress. In the current study, we demonstrate the ability of GS to defend lipid homeostasis imbalance and oxidative damage, and to restore hepatic function in ANIT-induced IC. Moreover, we demonstrate that GS exerts the above effect through the SIRT1/AMPK signaling pathway.

## 2. Materials and Methods

### 2.1. Reagents and Antibodies

*Panax ginseng* samples were obtained from Fusong Country Natural Biotechnology Co., Ltd. (Jilin, China). Doxorubicin hydrochloride (purity > 98%) was acquired from Kori Biotechnology Co., Ltd. (Wuhan, China). ANIT (purity > 98%) was obtained from Sigma (St. Louis, MO, USA). Bodipy was obtained from Invitrogen (Carlsbad, CA, USA) and DAPI was obtained from Sigma-Aldrich (St. Louis, MO, USA). Antibodies directed against AMPK, p-AMPK, Nrf2, HO-1, GCLC, NQO1, GCLM, and Sirt1 were obtained from Cell Signaling Technology (Beverly, MA, USA). Antibodies directed against SREBP-1, SCD-1, FAS, NTCP, HSL, CES1, PPARα, CYP7A1, CYP27A1, Mrp2, and β-actin were acquired from Absin Biochemical Company (Shanghai, China), and BSEP was acquired from Santa Cruz Biotechnology (Santa Cruz, CA, USA).

### 2.2. Preparation of GS and Fingerprint Detection

The extract from *Panax ginseng* roots was prepared as described in the literature [19]. The chemical profiles of ginsenosides were determined by high-performance liquid chromatography (HPLC) (purity > 80%) [19]. The chemical formula of GS is shown in Figure 1.

### 2.3. Animals and Experimental Protocol

C57BL/6J mice were acquired from the Experimental Animal Center of Tongji Hospital, Huazhong Science and Technology University (Wuhan, China). Animals were housed in a temperature- and humidity-controlled environment under a 12 h light–dark cycle and kept on a standard lab diet with water provided ad libitum. Animals were divided into seven groups randomly (*n* = 6) after 1 week of adjusted feeding. In brief, the normal control group, GS group (300 mg/kg), model (ANIT) group, ANIT+GS (30, 100 and 300 mg/kg) groups, and ANIT+UDCA (40 mg/kg) groups. Mice were pretreated with different doses of GS (oral gavage), vehicle (0.5% sodium carboxymethylcellulose), or UDCA for five consecutive days, during which the mice were treated with either ANIT (100 mg/kg) or vehicle (olive oil) by oral gavage on the third day. After ANIT was given at 48 h, all animals were sacrificed on the fifth day. Bile, blood, and livers were collected for subsequent study.

Hepatocyte-specific deletion of the SIRT1 gene (B6;129-Sirt1tm Ygu/J, also known as SirT1co, Jackson Stock no: 008041) mice (SIRT1^LKO^) were obtained from Jackson Stock Laboratory. SIRT1^LKO^ mice and wild-type mice were administered with GS by oral gavage (300 mg/kg) for five days and treated with ANIT (100 mg/kg) on the third day.

### 2.4. Cell Culture

HepG2 cells were obtained from ATCC (Manassas, VA, USA). Cells were grown to 80% confluency. After overnight incubation, ANIT (50 μM), ANIT (50 μM) with GS (1000 mg/mL), and ANIT (50 μM) with GS (1000 mg/mL) with doxorubicin hydrochloride (200 ng/mL) were added to the culture medium for 24 h. After 24 h, cells were harvested for follow-up experimental operation.

### 2.5. Detection of Biochemical Index

Markers of alanine aminotransferase (ALT), aspartate aminotransferase (AST), and hepatic function, etc., were assessed using commercial kids following the manufacturer’s directions (Nanjing Jiancheng Bioengineering Research Institute, Nanjing, China).

### 2.6. Histopathological Studies

Liver samples were preserved in 4% formaldehyde and then embedded in paraffin and sectioned (3–5 μm). Finally, they were stained with hematoxylin and eosin for pathological observation.

### 2.7. Oil Red O Staining

Oil Red O staining measures lipid droplet formation in hepatocytes. Frozen liver samples were counterstained with Mayer’s hematoxylin. Cells were then preserved in 4% formaldehyde and washed with 60% isopropanol after treatment, followed by staining with a working solution of Oil Red O at room temperature. Finally, cells were observed under the bright field of light microscopy.

### 2.8. Bodipy Staining

Experimental methods were referenced [20]. The HepG2 cells were seeded on coverslips in a 6-well plate, the cells then were stained with Bodipy and counterstained with DAPI.

### 2.9. Immunofluorescent Staining

Fluorescence stainings of SIRT1 protein, *p*-AMPK protein, ROS, and cell nuclei of livers were performed to understand the content changes in IC [21,22].

### 2.10. Detection of ROS in HepG2

The HepG2 cells were seeded to 6-well plates. Then, the concentrations of ROS in all groups were measured by using the ROS detection kit (SolelyBio, Beijing, China). The cells were treated with trypsin and then were harvested to be incubated with DCFH-DA [23].

### 2.11. Western Blotting Analysis

Liver tissues were homogenized and lysed in RIPA lysis buffer containing a protease inhibitor cocktail. Next, the supernatant was gathered and protein concentration was assessed with a BCA protein assay kit. Proteins were separated by 10% SDS-PAGE, transferred to a PVDF membrane, immunoblotted, blocked with 5% skim milk, washed with PBST three times, and incubated overnight at 4 °C with primary antibodies: anti-AMPK (1:1000), *p*-AMPK (1:1000), Nrf2 (1:1000), HO-1 (1:1000), GCLC (1:1000), NQO1 (1:1000), GCLM (1:1000), Sirt1 (1:1000), SREBP-1 (1:2000), SCD-1 (1:2000), FAS (1:2000), NTCP (1:2000), HSL (1:2000), CES1 (1:2000), PPARα (1:2000), CYP7A1 (1:2000), CYP27A1 (1:2000), Mrp2 (1:2000), β-actin (1:2000), and BSEP (1:2000). The next day, membranes were washed and incubated with secondary antibodies. Finally, protein bands were observed with ECL detection and calculated.

### 2.12. Quantification Real-Time PCR

Total RNA from mice livers was extracted using Trizol (Invitrogen Life Technology, Carlsbad, CA, USA). Next, total RNA was reverse transcribed into cDNA by PrimeScriptTM RT Marter Mix (Takara, Dalian, China). Gene expressions were quantified using SYBR PCR Master Mix (Takara, Dalian, China). Reports were analyzed by ABI StepOne Plus system (Applied Biosystems, Foster City, CA, USA). β-actin was applied as a standard to normalize the mRNA quantity. Specific primer sequences are shown in Appendix A.

### 2.13. RNA Silencing

HepG2 cells were grown to 80% confluency. Target genes were silenced according to previously established protocols [24]. After 48 h, cells were treated with ANIT (50 μM) and GS (1000 mg/mL) for 24 h, after which they were harvested for analysis.

### 2.14. Statistical Analysis

All the results were analyzed by GraphPad Prism 8.0 (GraphPad, La Jolla, CA, USA) and shown as the mean ± SD. The data were analyzed using Student’s *t*-test, and multiple comparisons were performed by ANOVA. Statistical significance was considered to be *p* < 0.05 or *p* < 0.01.

## 3. Results

### 3.1. GS Alleviated ANIT-Induced Cholestatic Liver Injury in Mice

As in our previous reports [19], GS remarkably alleviated the ANIT-induced decrease in body weight and increase in liver weight, as well as the ratio of liver/body weight in a dose-dependent manner (Figure 2A–C). The result of H&E staining showed that the ANIT-treated group developed swelling and the vacuolation of hepatocytes and hepatic inflammatory cell infiltration compared with the control group. Higher doses of GS showed better improvement than lower doses of GS (Figure 2D). In relation to the control group, the levels of serum AST, ALT, ALP, TBil, DBil, and TBA substantially rose in the ANIT-induced mice. The GS treatment significantly reduced the level of these serum markers in a dose-dependent manner (Figure 2E,F). To investigate the effect of the GS on the bile acid transporters and metabolic enzymes, PCR and Western blot analysis were applied to measure the bile-acid-related protein expression (Figure 3A,B). The GS (300 mg/kg) significantly promoted the expression of MRP2 at the gene and protein level in the ANIT-treated mice. Altogether, the GS treatment ameliorated the ANIT-induced hepatic liver injury.

### 3.2. GS Restored ANIT-Induced Lipid Homeostasis

To illustrate the role of the GS on lipid metabolism in the ANIT-treated mice, the levels of hepatic lipids and serum were observed. The treatment with the GS distinctly alleviated the increased levels of serum TC, TG, and LDL induced by ANIT, and enhanced the serum HDL levels. (Figure 4B). Lipid accumulation was likewise evaluated using Oil Red O staining. Control mice did not show steatosis, whereas the ANIT-treated mice exhibited a substantial rise in steatosis. The treatment with 300 mg/kg of GS markedly reduced fat droplet formation (Figure 4A). To further determine how the GS exerted hepatoprotective actions on reducing lipid deposit in the liver, we first ascertained the lipogenesis-related gene expression grades. The GS significantly reduced the ANIT-induced expression of the lipid biosynthesis gene SREBP-1, FAS, and SCD1. The GS also significantly enhanced the expression of the fatty acid β oxidation and lipolysis gene PPARα, HSL and CES1, counteracting the reduction in the expression of these genes following the ANIT treatment (Figure 4C). To provide further evidence of the above gene changes, we measured the protein expression, which was in line with the results observed of the PCR (Figure 4D). Together, these outcomes illustrate that the GS could alleviate the hepatic steatosis and lipid metabolism disorder through the modulation of gene expression following the ANIT-induced cholestatic liver injury.

### 3.3. GS Inhibited ANIT-Induced Oxidative Stress in Mice

ANIT generates prolonged oxidative stress, which accelerates the generation of ROS and reactive aldehydes [25]. Our current results also show that ANIT caused GSH depletion and a spectacular reduction of SOD, along with rise of MDA in liver tissues, compared with the control group. Interestingly, a marked reduction of MDA levels and restored hepatic SOD activity and GSH content were observed after pretreatment with GS (Figure 5A). After the ANIT treatment, we measured the fluorescent intensity of ROS in the liver tissues of mice. The GS treatment (300 mg/kg) significantly decreased the fluorescence intensity (Figure 5B). As Nrf2 is a vital endogenous antioxidant mediator, the role of GS on the Nrf2 pathway was tested. Compared with the ANIT group, the GS increased Nrf2 expression as well as the expression of its target genes HO-1, GCLM, GCLC, and NQO1 at the gene and protein level (Figure 6A,B). These results suggest that the pretreatment of GS reduced the ANIT-treated oxidative damage.

### 3.4. GS Ameliorated ANIT-Induced Low SIRT1/AMPK Expression in Mice

Research has found that SIRTI/AMPK expression plays a momentous role in glucolipid metabolism, and AMPK activation can decrease lipogenesis and increase fatty acid oxidation [26,27,28]. Consistent with the literature, we found that ANIT could cut down the levels of phosphorylated AMPK (p-AMPK) and SIRT1 in the liver in comparison with the control group. However, GS significantly increased the expression of proteins in the SIRT1/AMPK pathway (Figure 7A). Immunofluorescence assays further demonstrated these results (Figure 7B). These results collectively illustrated that GS activated the hepatic SIRT1/AMPK pathway.

### 3.5. GS Protected against ANIT-Induced IC by Activating AMPK/Nrf2 via SIRT1 Signaling Pathway In Vitro

To confirm the involvement of the SIRT1/AMPK pathway, we utilized the SIRT1 shRNA plasmids, as previously built [24], and doxorubicin hydrochloride (DH, AMPK inhibitor) so to eliminate the SIRT1 or AMPK expression in the HepG2 cells. The GS treatment improved the biochemical indexes (TG, TC, GSH, SOD) in the WT group, but the beneficial effect was inhibited by SIRT1 shRNA1 (Figure 8A,B). Further, the BODIPY and ROS staining results suggested that the GS treatment reduced the ROS and neutral lipid content based on the fluorescent density, but these changes did not occur after the SIRT1 shRNA1 intervention (Figure 8C,D).

Furthermore, the GS treatment significantly upregulated the expression of p-AMPK and Nrf2 in the ANIT-stimulated HepG2 cells. This was significantly reduced by SIRT1 shRNA1 transfection (Figure 8E). The HepG2 cells were pretreated with GS and/or DH (AMPK inhibitor, 200 ng/mL) before treatment with ANIT. The results indicate that exthe pression of SIRT1 was not affected after AMPK was inhibited (Figure 8F), indicating that AMPK is the downstream target gene of SIRT1.

These data showed that GS could protect from the ANIT-induced IC by improving the lipid metabolism and oxidative stress in a SIRT1/AMPK-dependent manner.

### 3.6. GS Ameliorated ANIT-Induced IC by Activating the AMPK/Nrf2 via SIRT1 in Mice

To further explore whether GS maintains lipid homeostasis and oxidative balance by acting on SIRT1, we used SIRT1^LKO^ mice. Compared with the ANIT+GS (WT), the liver weight was dramatically elevated in the ANIT+GS (SIRT1^LKO^) mice. Moreover, the GS had no restorative effects on the serum biomarkers, liver histopathology, and lipid accumulation in the SIRT1^LKO^ mice (Figure 9A–G). The protein results suggested that the GS could not raise the levels of SIRT1, p-AMPK, and Nrf2 in the context of SIRT1 knockout (Figure 10A,B).

These data indicated that GS could play a critical role in regulating the bile acid, lipid metabolism, and oxidative stress, and exerted this effect by the activation of the SIRT1/AMPK signaling pathway.

## 4. Discussion

Several liver diseases can build up on account of IC, including biliary atresia, principal sclerosing cholangitis, and chief biliary cirrhosis [29]. However, the pathological mechanism of IC is complex, and thus effective treatments are lacking. There have been many reports of GS used as a hepatoprotective treatment [30,31,32], but few reports in the field of cholestasis. Our previous studies have found that GS could improve IC in rats, although the molecular mechanism involved in treatment was unknown [19]. In the present study, we first found that GS could alleviate the lipid balance issues and oxidative stress in IC via the SIRT1/AMPK pathway, presenting a new pharmacological mechanism for treatment of IC.

As in the previous report [19], we found that GS could protect against liver parenchymal injury in the ANIT-induced IC mice, as well as reduce serum transaminase levels. The H&E staining revealed that GS (300 mg/kg) ameliorated inflammatory infiltration, improved structural damage, and recovered the morphology of the liver. Previous studies have found that GS could regulate expressions of the bile acid transporters BSEP and MRP2 [33], and our experiments confirmed this finding. Taken together, our results suggest that GS possessed a positive ameliorating action on ANIT-induced cholestasis.

IC has a complex pathological mechanism, with much attention focused on the metabolism and transport of bile acids. Less attention has been paid to other pathological processes associated with liver injury. Recent research has found that IC is accompanied by lipid metabolic disorders and oxidative stress, which are deemed to act distinctly in the pathological progress of IC [34]. In fact, clinical reports also have shown evidence of lipid disorders in IC patients [3,4]. However, the cause of lipid metabolism disorder in IC has not been elucidated. ANIT has been used as a hepatotoxic compound in rodents to induce intrahepatic cholestasis, mimicking the disease in humans [35]. Researchers have found that ANIT strongly altered the lipid profile by affecting the expressions of genes involved in lipid metabolism [36], making it a suitable model for use in this study. Since GS has been found to regulate the lipid metabolism and oxidative stress [30,31], we analyzed, for the first time, the role of GS on alleviating IC liver injury in these two areas and explored the mechanism involved.

In our studies, ANIT was used to induce steatosis in mice, with the data showing that a mass of lipid droplets were produced, accompanied by a significant rise in TG, TC, and LDL. The GS treatment reduced the amount of visible fat droplets and the level of TC and TG. ANIT increased ROS accumulation and the levels of MDA, while reducing hepatic GSH and SOD levels. Emerging evidence suggests that GS can lower lipid levels and attenuate oxidative stress, hypertrophy, inflammation, fibrosis, and apoptosis in cardiomyocytes, liver, and kidney [17,37]. Thus, we hypothesized that GS might regulate the lipid metabolism and oxidative stress-related gene expression in ANIT-induced IC. As predicted, the GS pretreatment significantly reversed these results, suggesting that GS can regulate the lipid metabolism and oxidant stress, and could be used to prevent IC.

Sirtuin1 (SIRT1) is a highly conservative protein NAD^+^-dependent deacylatelase, and thus its role is chiefly involved in the cellular metabolism [38]. As an important transmitter, SIRT1 participates in hepatic lipid metabolism and redox reaction homeostasis by mobilizing multiple signaling proteins, such as adenosine monophosphate-activated protein kinase (AMPK) or nuclear factor E2-related factor 2(Nrf2) [38]. As AMPK is an energy sensor protein, its activation plays a significant role in reducing lipid accumulation in the liver [39]. Following activation, AMPK can modulate hepatic energy metabolism through multiple mechanisms, including increased fatty acid oxidization, the suppression of lipid synthesis, as well as the repression of gluconeogenesis [40]. Nrf2 can increase the expression of antioxidant genes to protect against oxidative damage [41] and is a crucial modulator of the intracellular adaptive antioxidant response [42]. In some reports, GS has been shown to regulate SIRT1, AMPK, and Nrf2 [31,43,44]. Researchers found that SIRT1 activation improved metabolic markers to ameliorate cardio- and neuroprotection in both injured and noninjured subjects [43]. Other researchers discovered that ginsenoside Rb3-mediated cisplatin-induced nephrotoxicity alleviation was partly thanks to the regulation of AMPK/mTOR via the mediation of autophagy and the inhibition of apoptosis [44]. Moreover, Rb3 induced Nrf2 activation and upgraded the antioxidant pathway in the APAP-treated murine model of liver injury [31]. Our research was consistent with these findings and showed that GS efficiently increased SIRT1, phosphorylated AMPK and Nrf2, and reduced the expression of SREBP-1c, FAS, and SCD1 in ANIT-treated mice and HepG2 cells. Furthermore, GS markedly lessened lipogenesis genes and increased the expressions of genes associated with fatty acid oxidation, lipolysis genes, and antioxidation such as PPARα, HSL, CES1, and HO-1. We found that the beneficial effect of GS on the lipid metabolism and oxidative stress was due to the activation of the SIRT1 pathway, as the knockdown of SIRT1 in HepG2 cells and ANIT-treated mice eliminated the protective effect of GS. In addition, we proved that AMPK is a downstream protein of SIRT1, which mediates the GS-induced protection against ANIT-accelerated hepatic steatosis. Taken together, these outcomes suggest that SIRT1/AMPK signaling took part in the protective role of GS against ANIT-mediated lipid disorder and oxidative stress. Therefore, from the perspective of preclinical research, our results indicated the importance of GS in improving abnormal lipid metabolism and oxidative stress in the treatment of IC, which provides a promising drug for IC.

The GS, which includes a group of steroidal saponins composed of Rb, Rc, Re, Rg, etc., are responsible for the pharmacological effects of Panax ginseng [45]. The composition of GS is complex, and the material basis for its alleviation of cholestasis remains unclear. Recent research has found that Rc can activate SIRT1, leading to improved energy metabolism in cardiomyocytes and neurons [43], and Rg1 can recover carbon tetrachloride-induced acute liver injury by stimulating Nrf2 [31]. These studies suggested that we might be able to further elucidate the effective monomer of ginsenosides in the treatment of IC in the future.

## 5. Conclusions

In summary, we have demonstrated that GS attenuates ANIT-induced IC through the remission of lipid metabolism imbalance and oxidative stress, in which the activation of SIRT1/AMPK by GS is involved in its protective effects (Figure 11). This discovery suggests that GS could be a promising therapeutic agent for the treatment of cholestasis.

## Figures and Tables

**Figure 1 nutrients-14-03938-f001:**
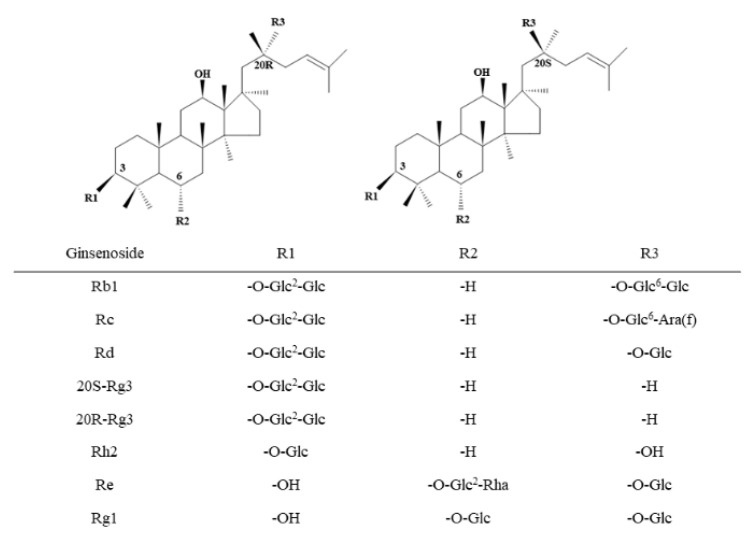
Structural formula of the ginsenoside (GS) monomer. The numerical superscript indicates the carbon on the glycosidic bond. Glc, glucose; Ara(f), arabinofuranose; Rha, rhamnose.

**Figure 2 nutrients-14-03938-f002:**
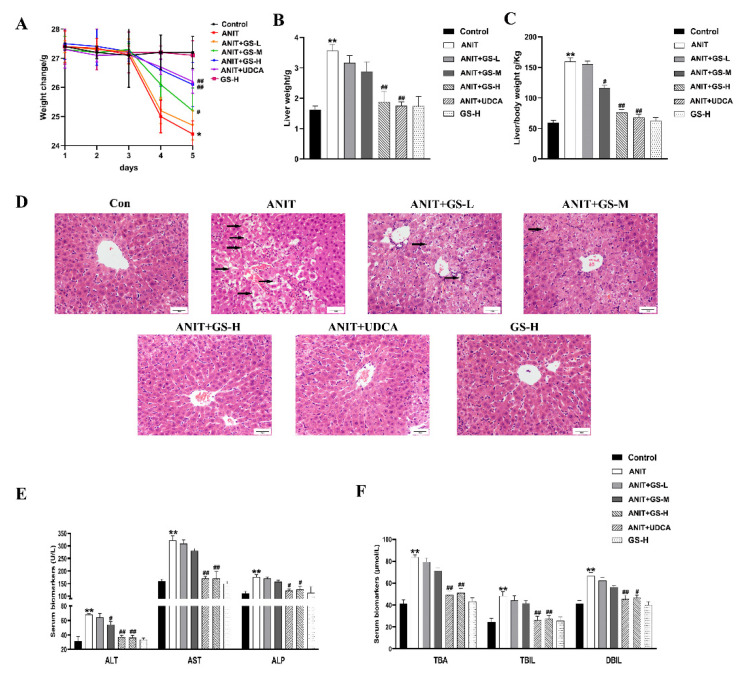
GS markedly improved cholestatic liver injury in ANIT-induced mice. (**A**) Body weight changes of mice from day 1–5. (**B**) Comparison of liver weight in each group on the fifth day. (**C**) Comparison of liver weight/body weight ratio in each group on the fifth day. (**D**) Histopathology in hematoxylin and eosin-stained liver sections (Scale bar, 50 um). Black arrows, histological damage. (**E**,**F**) The levels of ALT, AST, ALP, TBA, TBIL, and DBIL in serum of mice. Significantly different from control, ** *p* < 0.01, or ANIT, # *p* < 0.05 or ## *p* < 0.01.

**Figure 3 nutrients-14-03938-f003:**
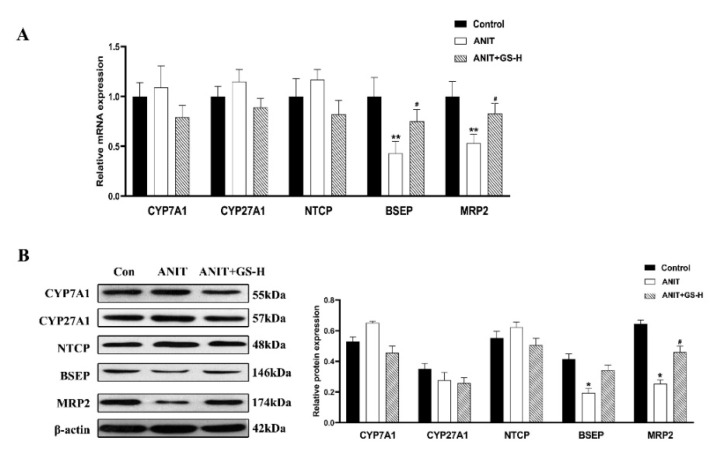
GS treatment improved expression of genes involved in bile acid homeostasis in the liver. (**A**,**B**) The mRNA levels and protein levels of bile metabolism-related genes (CYP7A1, CYP27A1, NTCP, BSEP, MRP2) in mice after ANIT and GS exposure. Significantly different from control, * *p* < 0.05 or ** *p* < 0.01, or ANIT, # *p* < 0.05.

**Figure 4 nutrients-14-03938-f004:**
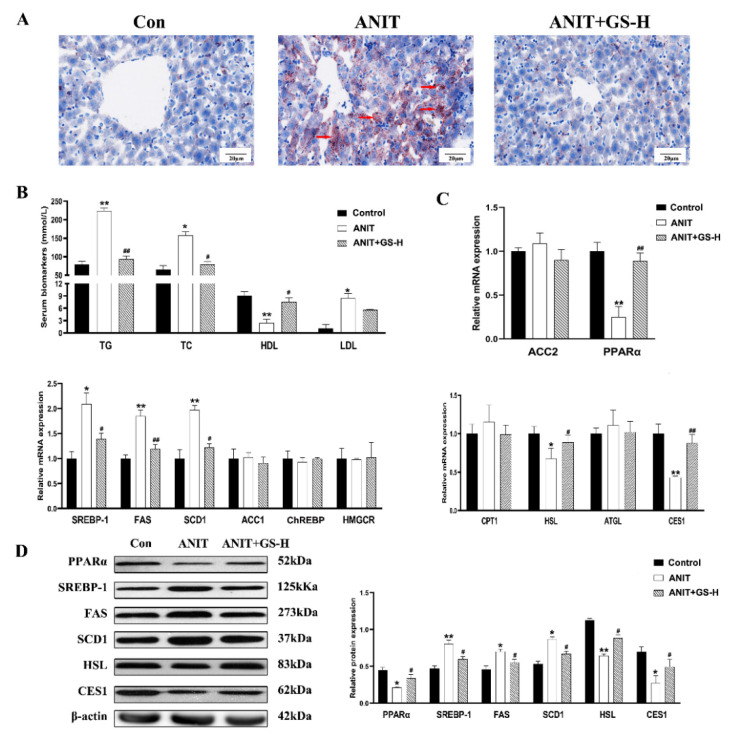
GS treatment improved fat metabolism gene profile in murine liver. (**A**) Morphology of frozen liver sections as tested by Oil Red O staining (scale bar, 20 μm). Red arrows, lipid droplets. (**B**) Levels of TG, TC, HDL, and LDL in serum of mice. (**C**,**D**) The levels of mRNAs (ACC2, PPARα, SREBP-1, FAS, SCD1, ACC1, ChREBP, HMGCR) and proteins (PPARα, SREBP-1, FAS, SCD1, HSL, CES1) of fat metabolism-related genes in mice after ANIT and GS exposure. Significantly different from control, * *p* < 0.05 or ** *p* < 0.01, or ANIT, # *p* < 0.05 or ## *p* < 0.01.

**Figure 5 nutrients-14-03938-f005:**
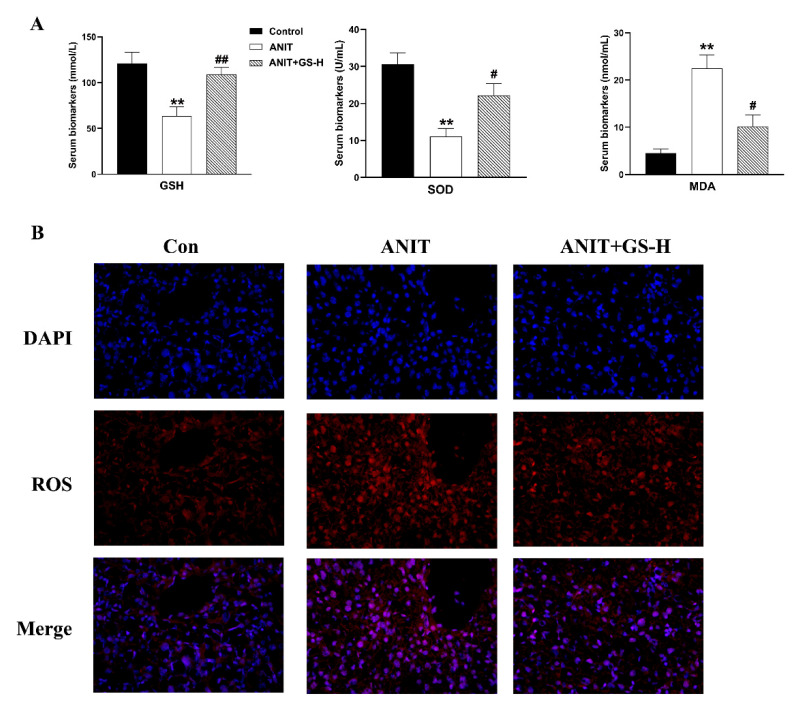
GS treatment improved oxidative damage in ANIT-induced mice. (**A**) The levels of GSH, SOD, and MDA in serum of mice. (**B**) Immunofluorescence stainings for ROS levels in liver tissue. Significantly different from control, ** *p* < 0.01, or ANIT, # *p* < 0.05 or ## *p* < 0.01.

**Figure 6 nutrients-14-03938-f006:**
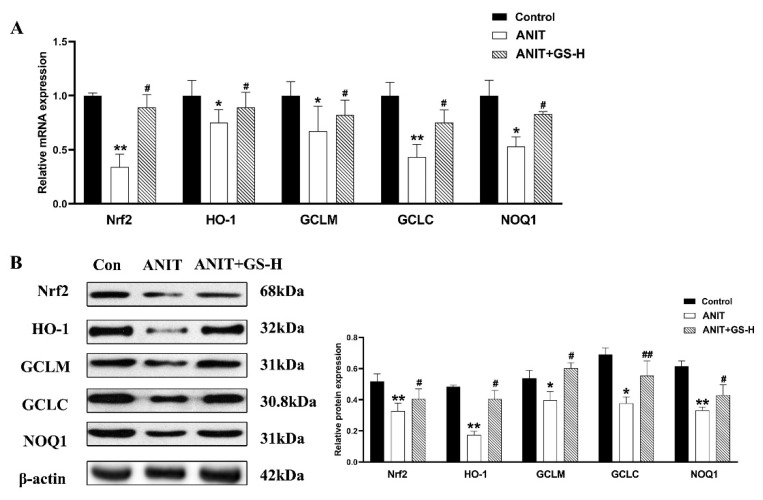
GS treatment improved the expression level of oxidative stress genes. (**A**,**B**) The mRNA and protein expressions of genes related to induce oxidative stress or associated genes (Nrf2, HO-1, GCLM, GCLC, and NQO1). Significantly different from control, * *p* < 0.05 or ** *p* < 0.01, or ANIT, # *p* < 0.05 or ## *p* < 0.01.

**Figure 7 nutrients-14-03938-f007:**
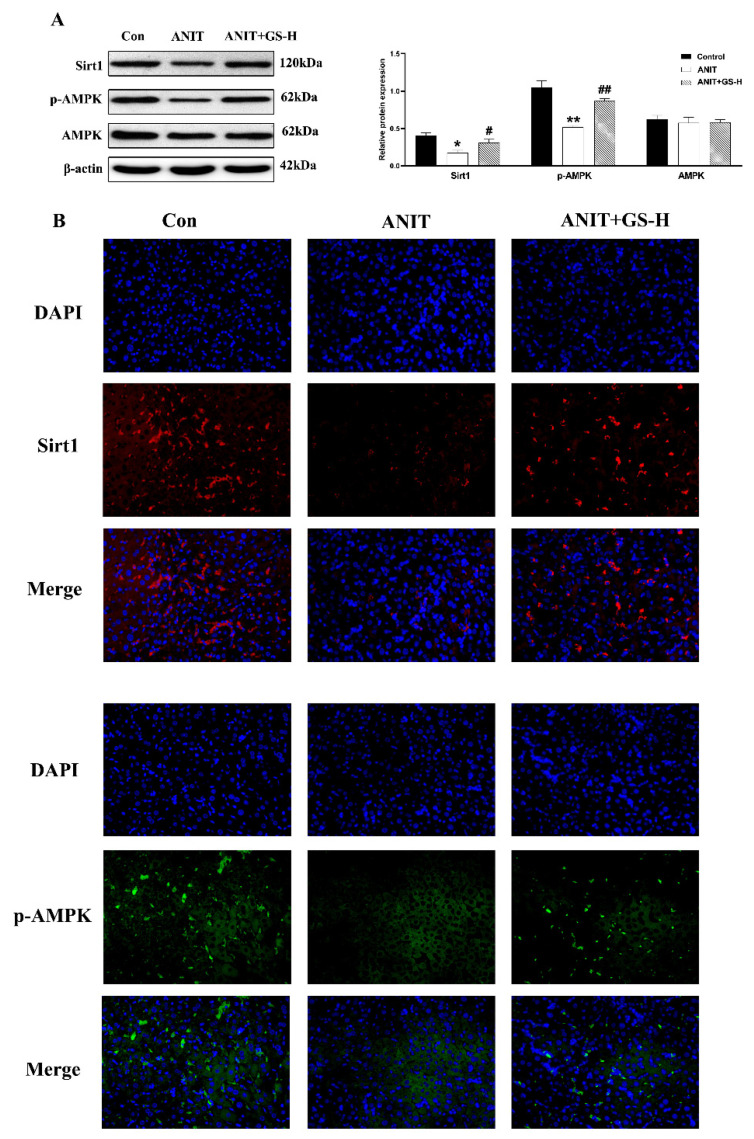
GS treatment improved the protein expression level of SIRT1/AMPK. (**A**) The protein expression levels of SIRT1, p-AMPK, and AMPK. (**B**) Immunofluorescence stainings for SIRT1 and p-AMPK protein levels in liver tissue. Significantly different from control, * *p* < 0.05 or ** *p* < 0.01, or ANIT, # *p* < 0.05 or ## *p* < 0.01.

**Figure 8 nutrients-14-03938-f008:**
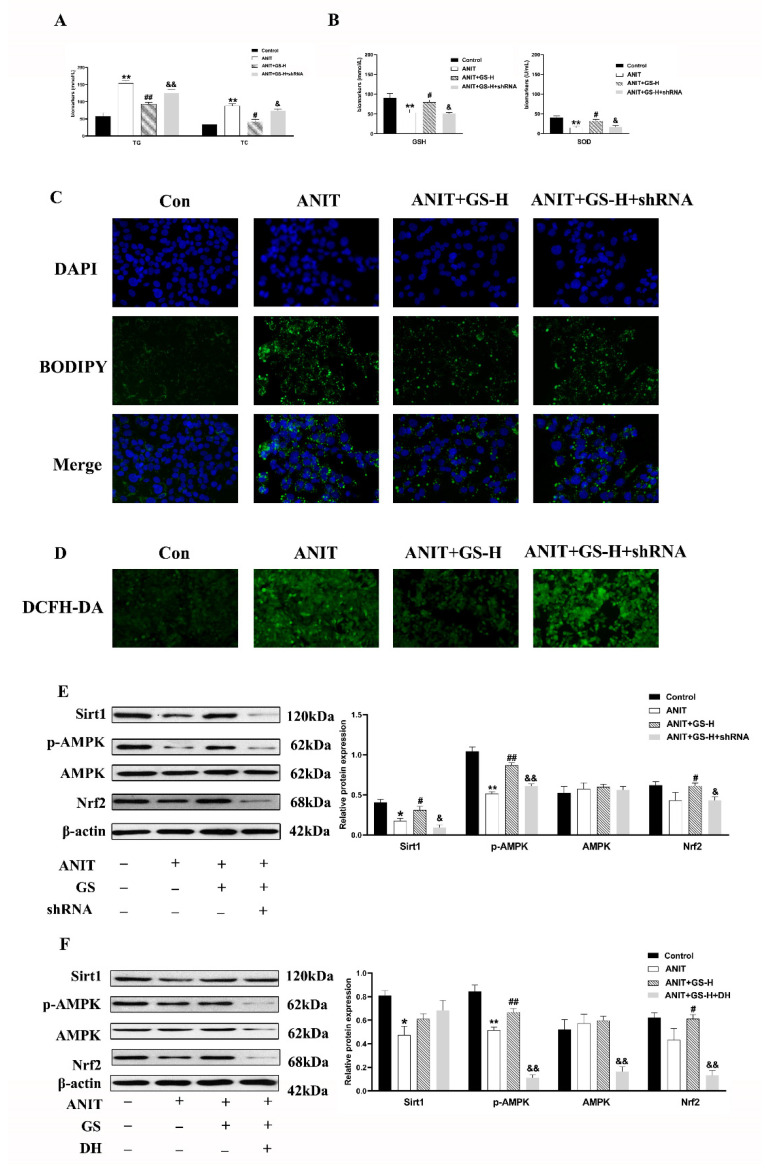
GS activated the SIRT1/AMPK signaling pathway in HepG2 cells. (**A**,**B**) Levels of TG, TC, GSH, and SOD in cell extracts. (**C**) Immunofluorescent analysis of neutral lipids accumulation (fat droplets) in HepG2 cells using BODIPY stain. (**D**) The production of ROS in HepG2 cells was assessed by DCHF-DA assay. (**E**,**F**) After the addition of sh-SIRT1 or doxorubicin hydrochloride (DH, AMPK inhibitor) the role of GS on SIRT1, p-AMPK, AMPK, and Nrf2 levels was tested by Western blot. Significantly different from control, * *p* < 0.05 or ** *p* < 0.01, or ANIT, # *p* < 0.05 or ## *p* < 0.01, or ANIT+GS-H, & *p* < 0.05 or && *p* < 0.01.

**Figure 9 nutrients-14-03938-f009:**
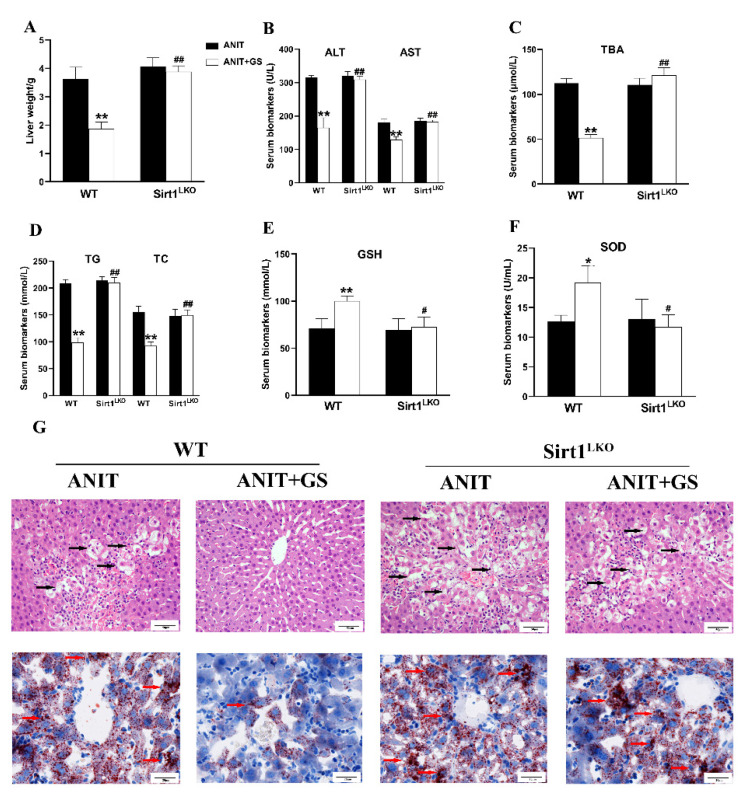
The therapeutic effect of GS was absent in SIRT1^LKO^ mice. (**A**) Liver weight changes (WT compared with SIRT1^LKO^. (**B**–**F**) The biochemical indexes of ALT, AST, TBA, TG, TC, GSH, and SOD in the serum of mice. (**G**) Histopathology in hematoxylin and eosin-stained liver sections (scale bar, 50 µm). Black arrows, histological damage. Morphology of frozen liver sections as tested by Oil Red O staining (scale bar, 20 µm). Red arrows, lipid droplets. Significantly different from ANIT (WT), * *p* < 0.05 or ** *p* < 0.01, or ANIT+GS (WT), # *p* < 0.05 or ## *p* < 0.01.

**Figure 10 nutrients-14-03938-f010:**
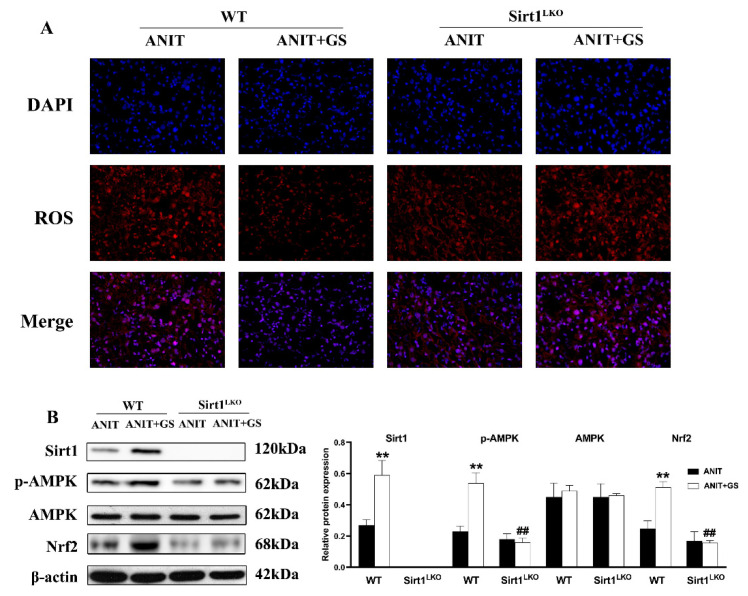
GS-induced activation of the SIRT1/AMPK signaling pathway was significantly reversed in SIRT1^LKO^ mice. (**A**) Immunofluorescent staining for ROS levels in liver tissue. (**B**) Protein levels of SIRT1, p-AMPK, AMPK, and Nrf2 levels in WT and SIRT1^LKO^ after GS treatment. Significantly different from ANIT (WT), ** *p* < 0.01, or ANIT+GS (WT), ## *p* < 0.01.

**Figure 11 nutrients-14-03938-f011:**
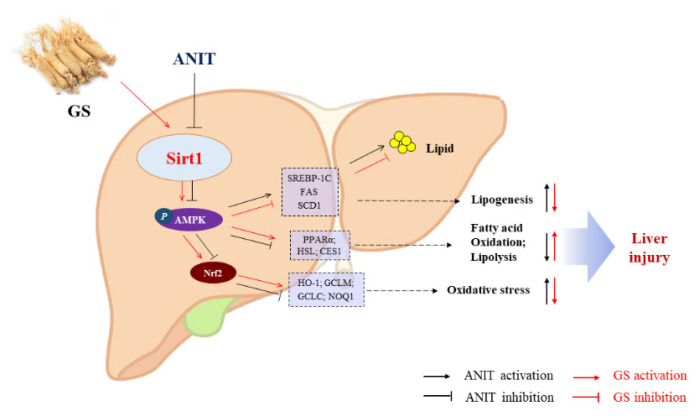
The proposed mechanism of GS-mediated protection in IC via the activation of the SIRT2/AMPK pathway. GS increases expression of SIRT1, leading to phosphorylation of AMPK and Nrf2, and the subsequent modulation of genes associated with inhibiting hepatic lipogenesis and increasing hepatic fatty acid oxidation and lipolysis.

## Data Availability

Data is contained within the article. All data were generated in-house, and no paper mill was used. All authors agree to be accountable for all aspects of work ensuring integrity and accuracy.

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
