# Peer review of "Ginsenosides Restore Lipid and Redox Homeostasis in Mice with Intrahepatic Cholestasis through SIRT1/AMPK Pathways"

_nutrients, 2022, doi:10.3390/nu14193938_

Round 1

Reviewer 1 Report

This paper investigated the effects of Ginsenosides on intrahepatic cholestasis in mice. This is an extensive and complete study. However, some minor problems need to be addressed: The text in Methods section 2.3 is very confusing. Was the GS intervention continued for five days before the ANIT administration? How long did the ANIT treatment last? How long did the entire experiment last? Was GS administered by oral gavage while ANIT administered in another way? Please clarify all these points in the manuscript.

Reviewer 2 Report

The authors describe their work on the possible protective effects of ginsenosides (GS) on lipid homeostasis disorder and oxidative stress in mice with alpha-naphthylisothiocyanate (ANIT)-induced intrahepatic cholestasis (IC) and to investigate the underlying mechanisms. It was found that GS can restore the balance of lipid metabolism and redox in the livers of ANIT-induced IC models via SIRT1/AMPK signaling pathway, thus exerting a protective effect against ANIT-induced cholestatic liver injury. It was concluded that the discovery suggested that GS could be a promising therapeutic agent for the treatment of cholestasis. This is an interesting study. Appropriate methodology has been employed and the manuscript is written very well. The authors are to be commended on the wealth of data generated. The conclusions appear to be justified based on the data at hand. I have some minor recommendations for consideration.

1.      Introduction. Although provided in the discussion section of the paper, can the authors provide a clear hypothesis to be tested in the introduction section of the paper?

2.      Results. Is it possible to identify areas of interest with arrows for histological images?

3.      Results. For some of the bar graphs, is it possible to increase the symbol size indicating statistical differences, as current size presents some difficulty to view.

4.      Discussion. Please describe the novelty aspect of the work.

5.      Discussion. The authors should elaborate and emphasize the clinical applicability and relevance of their findings.
